# Immune stealth-driven O2 serotype prevalence and potential for therapeutic antibodies against multidrug resistant *Klebsiella pneumoniae*

Meghan E. Pennini[1], Anna De Marco[2], Mark Pelletier[1], Jessica Bonnell[1], Romana Cvitkovic[1], Martina Beltramello[2], Elisabetta Cameroni[2], Siro Bianchi[2], Fabrizia Zatta[2], Wei Zhao[1], Xiaodong Xiao[1], Maria M. Camara[1], Antonio DiGiandomenico[1], Elena Semenova[1], Antonio Lanzavecchia[3], Paul Warrener[1], JoAnn Suzich[1], Qun Wang[1], Davide Corti [2] & C.Kendall Stover [1]

Emerging multidrug-resistant bacteria are a challenge for modern medicine, but how these pathogens are so successful is not fully understood. Robust antibacterial vaccines have prevented and reduced resistance suggesting a pivotal role for immunity in deterring antibiotic resistance. Here, we show the increased prevalence of *Klebsiella pneumoniae* lipopolysaccharide O2 serotype strains in all major drug resistance groups correlating with a paucity of anti-O2 antibodies in human B cell repertoires. We identify human monoclonal antibodies to O-antigens that are highly protective in mouse models of infection, even against heavily encapsulated strains. These antibodies, including a rare anti-O2 specific antibody, synergistically protect against drug-resistant strains in adjunctive therapy with meropenem, a standard-of-care antibiotic, confirming the importance of immune assistance in antibiotic therapy. These findings support an antibody-based immunotherapeutic strategy even for highly resistant *K. pneumoniae* infections, and underscore the effect humoral immunity has on evolving drug resistance.

[1] MedImmune, 1 Medimmune Way, Gaithersburg, MD 20878, USA. [2] Humabs BioMed SA, a subsidiary of Vir Biotechnology, Inc., Via Mirasole 1, 6500 Bellinzona, Switzerland. [3] Institute for Research in Biomedicine, Università della Svizzera Italliana, Via Vincenzo Vela 6, 6500 Bellinzona, Switzerland. Correspondence and requests for materials should be addressed to Q.W. (email: qunwang2001@hotmail.com) or to D.C. (email: davide.corti@humabs.ch) or to C.K.S. (email: stoverk@medimmune.com)

Among the most problematic multidrug-resistant (MDR) bacterial pathogens are the Gram-negative carbapenem-resistant enteric bacteria (CRE), including *Klebsiella pneumoniae*[1, 2]. *K. pneumoniae* colonizes the lower gastro-intestinal tract, from which it can disseminate, particularly when the commensal microbiota is disrupted by antibiotic treatment or when a patient is immunocompromised[3]. More than a third of *K. pneumoniae* clinical isolates express extended spectrum beta-lactamases (ESBL) and are resistant to third generation cephalosporins, aminoglycosides, tetracycline, and other antibiotics[4]. CRE isolates are additionally resistant to carbapenems and are strongly correlated with poor patient outcome[5, 6], with mortality rates as high as 60%.

While the majority of effort on understanding the spread of antibiotic resistance has focused on the acquisition of mutations or resistance genes that directly affect drug susceptibility, the most successful resistant strains clearly have additional adaptations. For example, the successful spread of the CRE sequence type 258 (ST258) clonal group is the subject of intense study because the diverse resistance genes this group carries are not unique to this lineage[7, 8]. In addition to bacterial factors, there is growing evidence demonstrating a link between host immunity and reduced antibiotic resistance. Since the onset of routine vaccinations against *Haemophilus influenzae*, *Streptococcus pneumoniae*, and *Neisseria meningitidis*, resistance within these organisms has been reduced or nearly eradicated[9]. Effective immunity can decrease the severity and/or duration of an infection thus lessening the need for antibiotic intervention and reducing the selective pressure imparted by antibiotic exposure. Host humoral immunity can also work in concert with antibiotic therapy to synergistically clear drug-resistant pathogens that cannot be cleared with antibiotic alone[10, 11].

Lipopolysaccharide (LPS) is a critical component of the Gram-negative bacterial outer membrane. The earliest antibody-based strategies against Gram-negative pathogens targeted the conserved Lipid A component of the LPS inner core with the aim of neutralizing lipid A proinflammatory properties, but these efforts failed for a variety of reasons[12–15]. Beyond the conserved inner core, there are at least seven distinct Klebsiella LPS serotypes defined by unique O-antigen structures[16]. *K. pneumoniae* O1 and O2 LPS O-antigens share the D-galactan I disaccharide unit, but O1 also includes the D-galactan II disaccharide and is therefore more complex in structure and larger in size[17, 18]. In previous target agnostic efforts to identify antibodies against *K. pneumoniae*[19], we found that LPS serotype-specific antibodies were the most potent in promoting bacterial killing in vitro.

In this study, we survey a large collection of clinical isolates to determine the relative prevalence of LPS serotypes, particularly in multidrug-resistant isolates. We find that LPS O2 antigen serotype has increased two to threefold in prevalence in both *K. pneumoniae* ESBL and CRE multidrug-resistant strain groups relative to the susceptible group. This finding is surprising as strains expressing the O2-antigen, including ST258 strains, are more sensitive to human serum killing than the related O1 antigen expressing strains. We demonstrate ST258 strains are almost exclusively the O2 serotype, but the overall increased O2 prevalence is not limited to ST258. These data suggest a selective advantage for the LPS O2-antigen type in the context of antibiotic pressure. We find that O2 strain prevalence may be explained by its reduced immunogenicity that imparts a stealth advantage against antibody-driven mechanisms of clearance. In addition, we identify serotype-specific anti-O1 and anti-O2 human monoclonal antibodies (mAbs) that exhibit potent opsonophagocytic killing of *K. pneumoniae* strains and provide protection in mouse models of infection. These data indicate that protective non-capsular

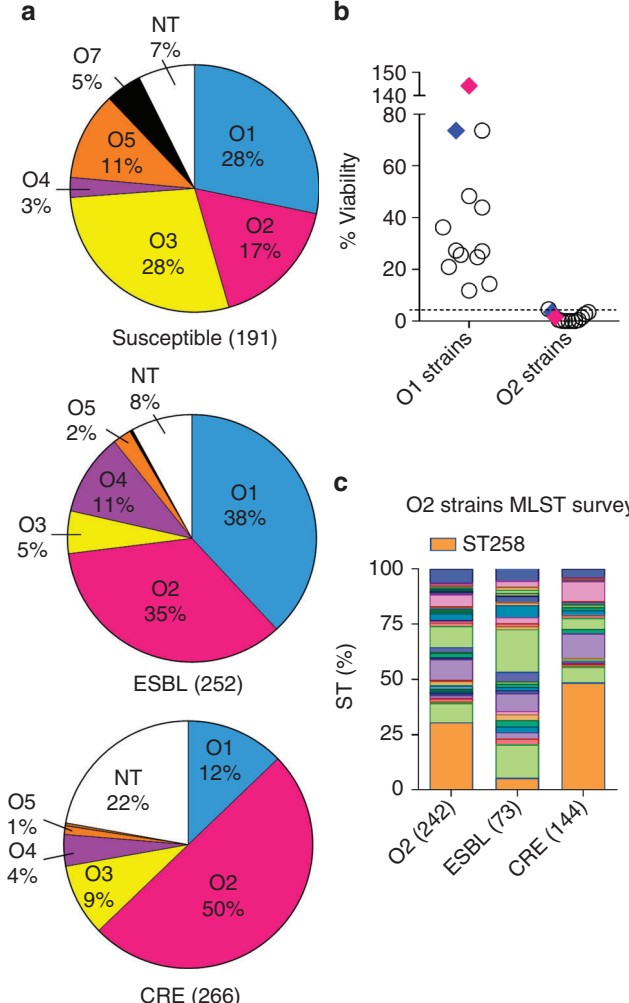

**Fig. 1** *Klebsiella pneumoniae* serotype O2 prevalence in MDR strains. **a** LPS serotype-specific antibodies were used to determine the serotype of 709 *K. pneumoniae* clinical isolates from 38 different countries by western blot. Pie charts represent the distribution of each serotype as a percentage of the total in each antibiotic resistant group. Susceptible, ESBL and CRE strains are categorized based on their MIC profile as described in Methods. More information about the source of the isolates is listed in Supplementary Table 1. **b** Serum sensitivity assays were performed in 45% pooled human sera using a panel of 12 O1 isolates, 12 O2 isolates and two separate isogenic pairs genetically converting an O2 strain to O1 (Kp961842, magenta diamonds) or an O1 strain to O2 (Kp1131115, blue diamonds). Percent viability is the ratio between CFUs present in buffer control media and CFUs present in human sera after 2 h growth. **c** A subset of serotype O2 isolates was further analyzed by Sanger or next generation sequencing to determine ST types. The CRE ST258 sequence type is represented by the shaded orange portion of each bar to highlight the percent of ST258 isolates observed in each category

mAbs targeting less diverse anti-O antigen types expressed by MDR strains can be identified. Importantly, these serotype-specific O1 and O2 antibodies synergistically augment anti-biotic therapy even against drug-resistant strains, further supporting an important role for antibody-mediated immunity in the context of antibiotic therapy. Together, these findings support that O-antigen immunogenic stealth provides a selective advantage to drive multidrug resistance and that protective antibodies can synergize with antibiotic therapy, thus underscoring the importance of humoral immunity to drug-resistant bacteria.

## Results

**O2 LPS serotype prevalence in multidrug-resistant strains**. We surveyed O-antigen serotype prevalence in a geographically diverse strain collection consisting of 709 clinical isolates collected between 2012 and 2014, from 162 different hospitals in 38 countries, spanning six continents and from a variety of body infection sites (Fig. 1a, Supplementary Table 1). Isolates were intentionally selected to represent comparable numbers of MDR (both ESBL and CRE) and fully susceptible isolates to generate the first large-scale analysis comparing serotype distribution as a function of antibiotic resistance. The serotype distribution observed among antibiotic susceptible strains was largely in line with serotype frequencies previously determined in the 1990s, including the predominance of the O1 serotype[20, 21]. However, with the development of ESBL and later emergence of CRE strains in the last two decades, O2 serotype appears to be the dominant serotype within both of these distinct MDR bacterial populations in comparison to the susceptible strain population (Fig. 1a). This result is unexpected in light of previous findings that O1 strains are much more resistant to complement dependent killing than other O serotypes[22–25]. Therefore, we evaluated the relative serum susceptibility of randomly selected O1 and O2 strains from the above clinical isolate collection to determine if O2 strains were resistant to human serum. However, we found O2 strains were much more sensitive to human serum killing than strains expressing the related O1 serotype (>2000 times more sensitive on average, Fig. 1b). Furthermore, isogenic conversion of an O1 strain (Kp1131115) to O2 by deletion of the *wbbYZ* locus resulted in a dramatic 21-fold decrease in serum resistance, and reciprocal conversion of an O2 strain (Kp961842) to O1 by the addition of these genes resulted in a 90-fold increase in serum resistance (Fig. 1b).

One explanation for the dominance of the O2 serotype within the MDR population could be an association with clonal outbreaks, such as the pervasive CRE Klebsiella global "super clone" ST258[31–33]. While we found the vast majority (85%) of ST258 isolates to be of the O2 serotype, only 49% of the CRE O2 isolates and just 5% of ESBL O2 isolates typed as ST258 (Fig. 1c).

Therefore, simple clonal expansion does not wholly explain the O2 prevalence in MDR strains, particularly within the ESBL subset. While Klebsiella carbapenem resistance is most often attributed to the acquisition of plasmid encoded *kpc* genes, other genes such as *oxa* and the emerging *ndm*[34, 35] are also of concern. Interestingly, O2 was also the predominant serotype in isolates that carry any of these genes (Supplementary Fig. 1b). These data suggest therapeutics targeted against O2 LPS could provide better coverage against MDR isolates than those specifically targeting only a major sequence type such as ST258.

**Klebsiella O2 LPS is less immunogenic than O1 LPS**. O2 MDR prevalence suggests an unrecognized selective advantage for these strains, particularly since it was confirmed that O2 strains are markedly more serum susceptible and because this increased prevalence is not entirely attributable to the success of a particular clone. While identifying antibodies for our serotype screen, we observed a much lower frequency of O2 specific B cells compared to O1 or O1/O2 cross reactive B cells, possibly due to a lower immunogenicity of the O2-antigen. To address this, human immune responses to O1 or O2 LPS were determined using plasma samples from 103 healthy donors and 6 O2 KP infected ICU patients. Titers indicated a dominant IgG response to O1 LPS compared to O2 (Fig. 2a, c, d). Of note, 5 out of 6 ICU plasma samples had anti-O2 LPS IgG antibodies, but those titers were paralleled by high titers of anti-O1 antibodies suggesting a significant fraction cross-react with O1 LPS (Fig. 2a). We also performed a clonal analysis using in vitro polyclonal stimulation, which selectively activates memory B cells in unfractionated tonsillar lymphocytes (antigen-specific memory B cell repertoire analysis, AMBRA[36]), (Fig. 2b–d, Supplementary Fig. 2). The frequency of LPS reactive memory B cells was higher in the IgM repertoire as compared to the IgG repertoire (Fig. 2c, d), consistent with the concept that the IgM B cell memory repertoire is dominated by anti-polysaccharide antibodies. Interestingly, the frequency of O2-specific memory B cells, both IgM and IgG, was significantly lower as compared to O1 specific. Together, these results suggest that O2 specific antibody responses may be

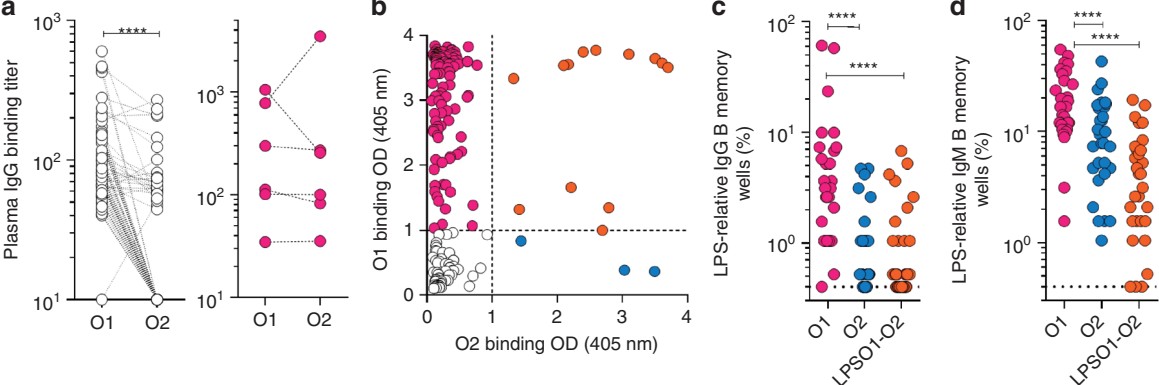

**Fig. 2** Analysis of the human antibody response to *K. pneumoniae* O1 and O2 LPS. **a** O1 and O2-specific IgG antibodies were measured by ELISA in plasma derived from 103 healthy donors (left panel) and 6 ICU patients (right panel) diagnosed with O2 *K. pneumoniae* infection. Shown are the endpoint titers based on serial dilutions. Lines between data points indicate data acquired from the same sample. **b–d** Analysis of antigen-specific repertoires (AMBRA, antigen-specific memory B cell repertoire analysis) from tonsillar IgG and IgM memory B cells of 33 donors (30 of the 33 were available for IgM analysis, Supplementary Fig. 2). Total tonsillar lymphocytes were plated and stimulated with R848 and IL-2 and the 10-day culture supernatants (192 cultures per donor) were analyzed by ELISA for the presence of IgG or IgM antibodies that bind to O1 LPS, O2 LPS or uncoated control plates. The OD values obtained from the analysis of uncoated plates were subtracted from the analysis of the O1 and O2 specific responses. **b** Shown are ELISA OD values of individual cultures from the analysis of one representative donor. Cut-off values are indicated by a dotted line. O1, O2 and O1/O2 reactive cultures are represented as magenta, blue and orange circles, respectively. **c, d** Each data point represents the frequency of the cultures (n = 192) scoring as positive for IgG (**c**) or IgM (**d**) reactivity to O1 or O2 LPS for each of the donors analyzed, according to the cut-off values shown in **b**. Data was analyzed using the Wilcoxon matched-pair signed rank test to assess significance. ****p < 0.0001

disproportionately lower than O1 or O1/O2 cross-reactive antibody responses.

To further investigate the immunogenicity of O1 and O2, mice and rabbits were immunized with purified O1 or O2 LPS. While O1 LPS-immunized animals had high titers against O1 LPS, O2 LPS-immunized animals had very low titers against O2 LPS (Fig. 3a, b). The poor antibody response to O2 LPS in animals is consistent with the human data. Mice were also immunized with either member of an isogenic O1/O2 bacteria pair. Mice

immunized with the O1-expressing wild-type (WT) strain generated high titers of antibodies against O1 LPS, while those immunized with the isogenic O2-expressing ΔwbbYZ strain did not generate a detectable response towards O2 LPS (Fig. 3c). Either immunization generated equivalent titers to outer membrane protein A (OmpA) and to a total carbohydrate extract (Fig. 3d). Additionally, both LPS serotypes stimulated a NF-κB-driven luciferase reporter gene to the same degree, indicating that O2 LPS is not defective in stimulating TLR4-driven inflammatory

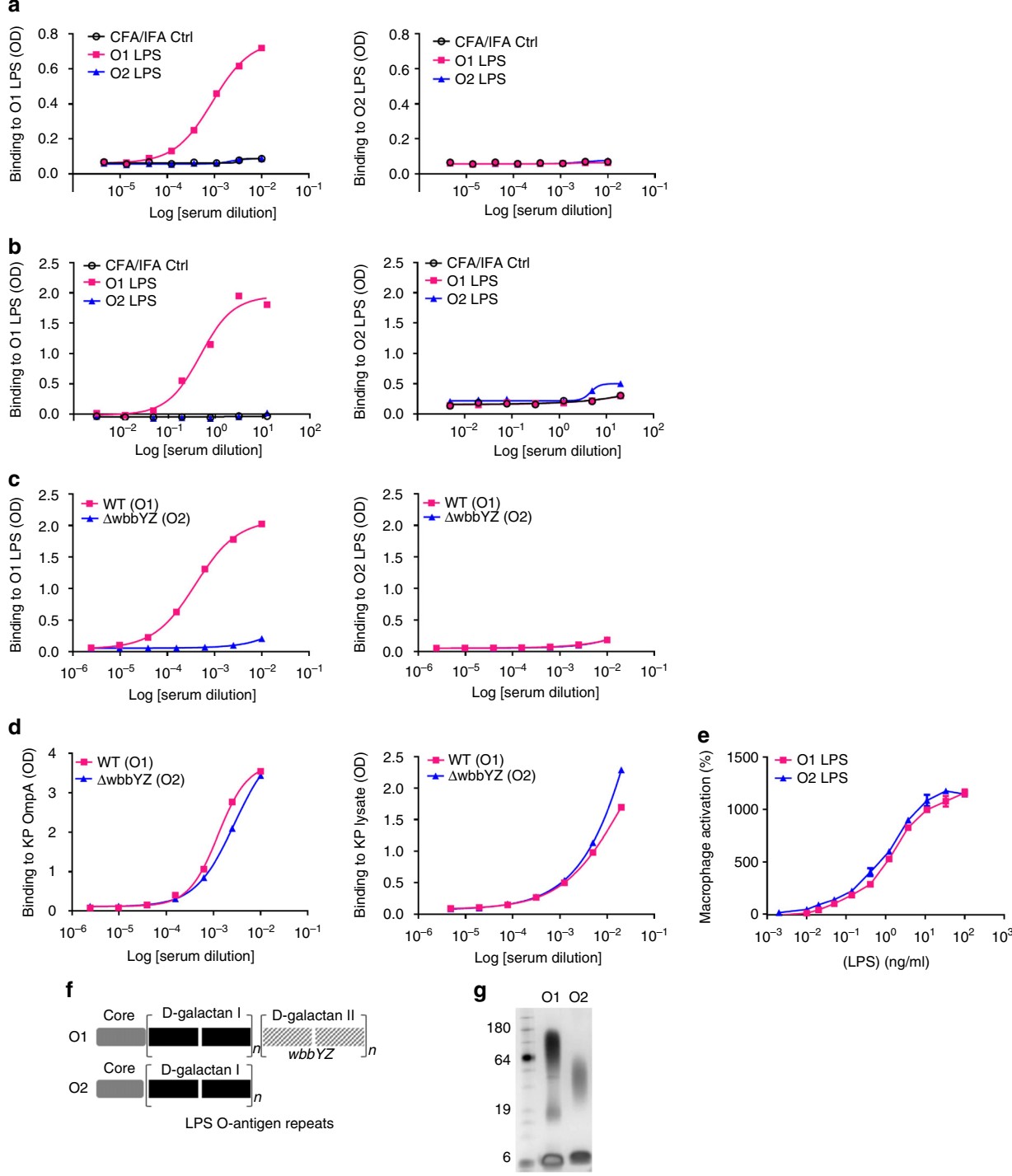

responses (Fig. 3e). O2 LPS is less complex in structure and has a lower molecular weight than O1 LPS (Fig. 3f, g) which may contribute to its reduced antigenicity, consistent with prevailing findings that T-cell-independent type 2 antigens require larger molecular weight, repetitive antigenic epitopes to effectively cross-link B-cell receptors to stimulate antibody responses[24, 37, 38]. Together, these data illustrate the weak antigenicity of O2-antigen and suggest that less antibody-mediated protection against O2 strains may contribute to their higher prevalence, particularly among recently emerging MDR strains against which antibiotic therapy is marginal.

**O1 and O2 LPS specific mAbs mediate opsonophagocytic killing.** In an effort to find functional protective mAbs, human peripheral blood and tonsil memory B cells were next isolated based on reactivity against purified O1 and O2 LPS, the most prevalent serotypes found in clinical isolates (Fig. 1a). All anti-LPS mAbs isolated were IgG2, consistent with the concept that this is the predominant IgG subclass produced in response to polysaccharides, but were converted and recombinantly expressed as human IgG1 for functional analysis. Several O1/O2 cross-reactive antibodies were identified and exemplified by mAb KPN70 which exhibited very high binding affinity to both O1 (~KD = 0.18 nM) and O2 (~KD = 0.06 nM) LPS, likely recognizing the shared O1/O2 polysaccharide component D-galactan I (Fig. 4a). Similarly, anti-O1 mAbs with strong binding specificity were readily identified and exemplified by mAb KPE33 (~KD = 1.8 nM). Consistent with the above data, O2 specific mAbs were exceedingly rare. Nevertheless, mAb KPN42 was isolated and determined to be highly specific for O2 LPS (~KD = 4.3 nM). Further binding analysis on O1 LPS showed no competition between KPE33 and KPN70, indicating that they recognize different epitopes on O1 LPS (Supplementary Fig. 3). Similarly, KPN42 did not block KPN70 binding to O2 LPS, however, the higher affinity mAb KPN70 partially blocked the binding of KPN42. This uni-directional blocking might be due to allosteric or steric hindrance caused by the binding of a high affinity mAb (in this case KPN70) near the binding region of a lower affinity mAb (KPN42). Overall, these binding data suggest recognition of distinct, albeit proximal, epitopes on O2 LPS by mAbs KPN70 and KPN42, and on O1 LPS by mAbs KPN70 and KPE33. Binding to whole bacteria was also analyzed by confocal microscopy. KPE33 and KPN42 uniformly bound the bacterial surface of encapsulated O1 and O2 strains, respectively, while the cross-reactive mAb KPN70 displayed a similar uniform pattern on the O2 strain, but exhibited a punctuate staining pattern on the O1 strain (Fig. 4b). The reduced binding of KPN70 to the surface of O1 strains compared to purified O1 LPS is likely due to limited accessibility of this epitope in the context of O1 vs. O2 whole bacteria. In support of this hypothesis, KPN70 bound to O2 but not O1 strains as measured by flow cytometry (Supplementary

Fig. 6a). In addition, strains that were converted from O1 to O2 showed enhanced KPN70 binding, and strains that were converted from O2 to O1 had reduced binding (Supplementary Fig. 6b). O-antigen was confirmed to be the target antigen of each of the mAbs (KPE33, KPN42 and KPN70) as they do not bind to mutant strains deficient in O-antigen expression (ΔwecA mutants, Supplementary Fig. 6c).

Additional assays were performed to assess the functional activity of the mAbs. LPS neutralization assays demonstrated that both O1 and O2 LPS were potent stimulators of NF-κB driven expression of luciferase. Interestingly, only KPN70 exhibited LPS neutralization activity (Fig. 4c), indicating that antibody binding to O-antigen does not necessarily result in blocking LPS activation of host target cells. Next, the mAbs were assessed for opsonophagocytic killing activity against luminescent wild-type and isogenic capsule-deficient K. pneumoniae strains. None of the mAbs mediated killing of wild-type encapsulated strains (Supplementary Fig. 4). When capsule-deficient mutant strains were used, the O1-specific mAb KPE33 mediated dose-dependent killing of the O1, but not the O2, target strain (Fig. 4d). Conversely, the O2-specific mAb KPN42 effectively mediated killing of the O2 strain only. Interestingly, the O1/O2 cross-reactive mAb KPN70 mediated opsonophagocytic killing activity only against the O2, but not the O1, target strain, possibly due to the lack of uniform binding on the bacterial cell surface as seen by confocal imaging (Fig. 4b).

**Anti-O-antigen in vivo protection is serotype specific.** Given their opsonophagocytic killing of two prominent LPS serotypes (summarized in Fig. 4e), we tested the protective potential of these mAbs in murine models of K. pneumoniae infection against wild-type encapsulated strains. Since each mAb targets a unique LPS antigen (O1 or O2), murine models were developed independently as a mimic of various clinical K. pneumoniae infections. The serotype-specific anti-O1 (KPE33) and anti-O2 (KPN42) mAbs significantly improved survival outcomes compared to the negative control mAb when used to treat infection with an O1 or O2 strain, respectively, in acute murine pneumonia (Fig. 5a) and bacteremia (Fig. 5b) models. In the bacteremia model, mortality is more rapid with a lower bacterial challenge dose than the pneumonia model resulting in a narrower therapeutic window. As such, the mAbs were most effective when administered prophylactically (Fig. 5b) vs. in treatment (Supplementary Fig. 5b), though either mAb delivery was significantly protective. Despite its high affinity binding to both O1 and O2 LPS and its potent LPS neutralizing activity, the anti-O1/O2 cross-reactive mAb KPN70 failed to improve survival outcome in the murine pneumonia model with an O1 strain and only moderately improved survival with an O2 strain (Fig. 5a). Interestingly, KPN70 had some activity against both an O1 strain and O2 strain in bacteremia (Fig. 5b, Supplementary Fig. 5b). The

**Fig. 3** K. pneumoniae O2 LPS is less immunogenic than K. pneumoniae O1 LPS. Purified O1 and O2 LPS were used for immunization of mice (**a**) or rabbits (**b**) with 0.5 mg LPS per animal. Immune sera were collected from O1 and O2 immunized animals and used to assess antibody titer against KP O1 or O2 LPS by ELISA. Sera from mock immunized animals with adjuvant only (black open circle) were used as a negative control. **c** Mice were immunized subcutaneously with either heat killed Kp1131115 (WT, an O1 strain) or the isogenic matched Kp1131115ΔwbbYZ (ΔwbbYZ, O2 serotype). Sera were collected 7 days after the final injection and used to assess antibody titer against K. pneumoniae O1 or O2 LPS by ELISA. **d** Sera collected in **c** were used to assess antibody binding to a protein target (OmpA) and to proteinase K bacterial lysate by ELISA. **e** Immune activation of cells in vitro was compared for purified O1 and O2 LPS. RAW264.7 cells stably transfected with a NF-κB driven luciferase promoter were stimulated with various doses of K. pneumoniae O1 or K. pneumoniae O2 LPS for 2 h. Percent activation was calculated as the amount of luminescence in stimulated cells vs. non-stimulated. Error bars indicate s.d. of each data point. **f** A schematic representation of KP O1 and O2 LPS. The gray region represents the membrane proximal LPS core region, the black region represents the D-galactan I (D-gal I) repeating O antigen sugars, the hatched gray filled region represents the O1 specific D-galactan II (D-gal II) repeating O antigen sugars encoded by the wbbYZ locus. Deletion of this locus converted the O1 strain (Kp1131115 WT) to an O2 expressing strain (Kp1131115ΔwbbYZ). **g** Silver stain of a SDS-PAGE gel showing the relative molecular weight of K. pneumoniae O1 LPS (lane 2) vs. K. pneumoniae O2 LPS (lane 3)

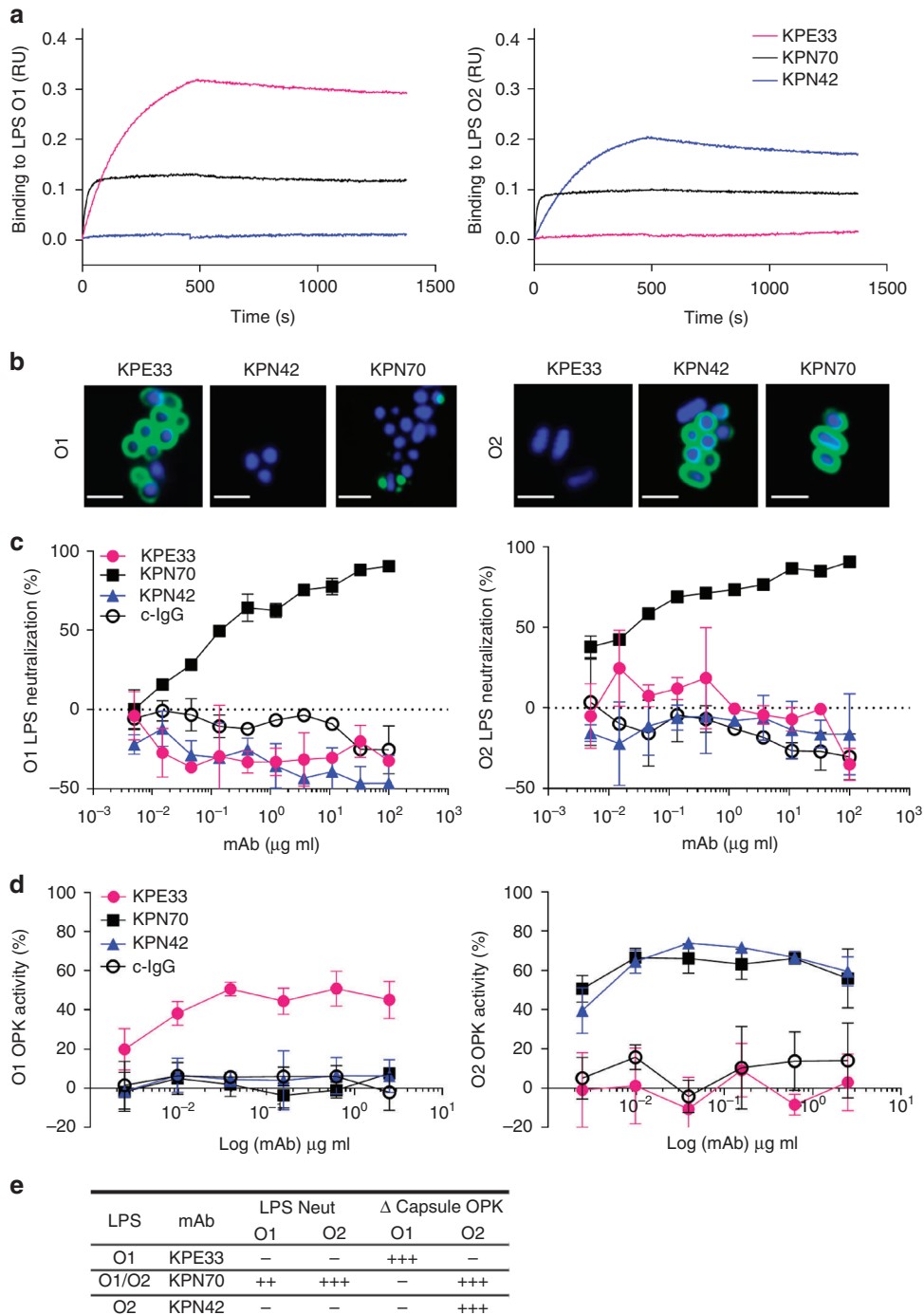

**Fig. 4** In vitro characterization of anti-O antigen mAbs. **a** Association/dissociation curves of *K. pneumoniae* O1 or *K. pneumoniae* O2 LPS binding to mAbs KPE33, KPN70, and KPN42 were measured by ForteBio Octet using protein A capture probe pre-loaded with each mAb. **b** Confocal images of mAb binding to an O1 (Kp43816) or O2 (Kp9148) strain (scale bar, 2 μm). **c** LPS neutralization was measured using RAW264.7 cells transfected with a NK-kB driven luciferase promoter stimulated with 2 ng ml$^{-1}$ of purified O1 or O2 LPS in the presence of KPE33, KPN70, KPN42 or the isotype-matched control mAb (c-IgG). Percent inhibition was calculated as the amount of luminescence (RLU) in treated wells divided by the RLU in wells stimulated with LPS alone multiplied by 100. **d** Opsonophagocytic killing assays were performed using various concentrations of mAbs in the presence of the phagocytic cell line HL-60. The capsule-deficient luminescent mutants Kp43816ΔcpsB and Kp8570ΔcpsB were used as the O1 and O2 serotype targets, respectively. Percent killing was calculated based on the amount of luminescence after 2.5 h incubation in treated wells vs. wells with bacteria only. An isotype-matched mAb (c-IgG) is used as a negative control. **e** Summary of the in vitro functional activities against O1 and O2 *K. pneumoniae* for each mAb. All error bars indicate s.d. at each data point

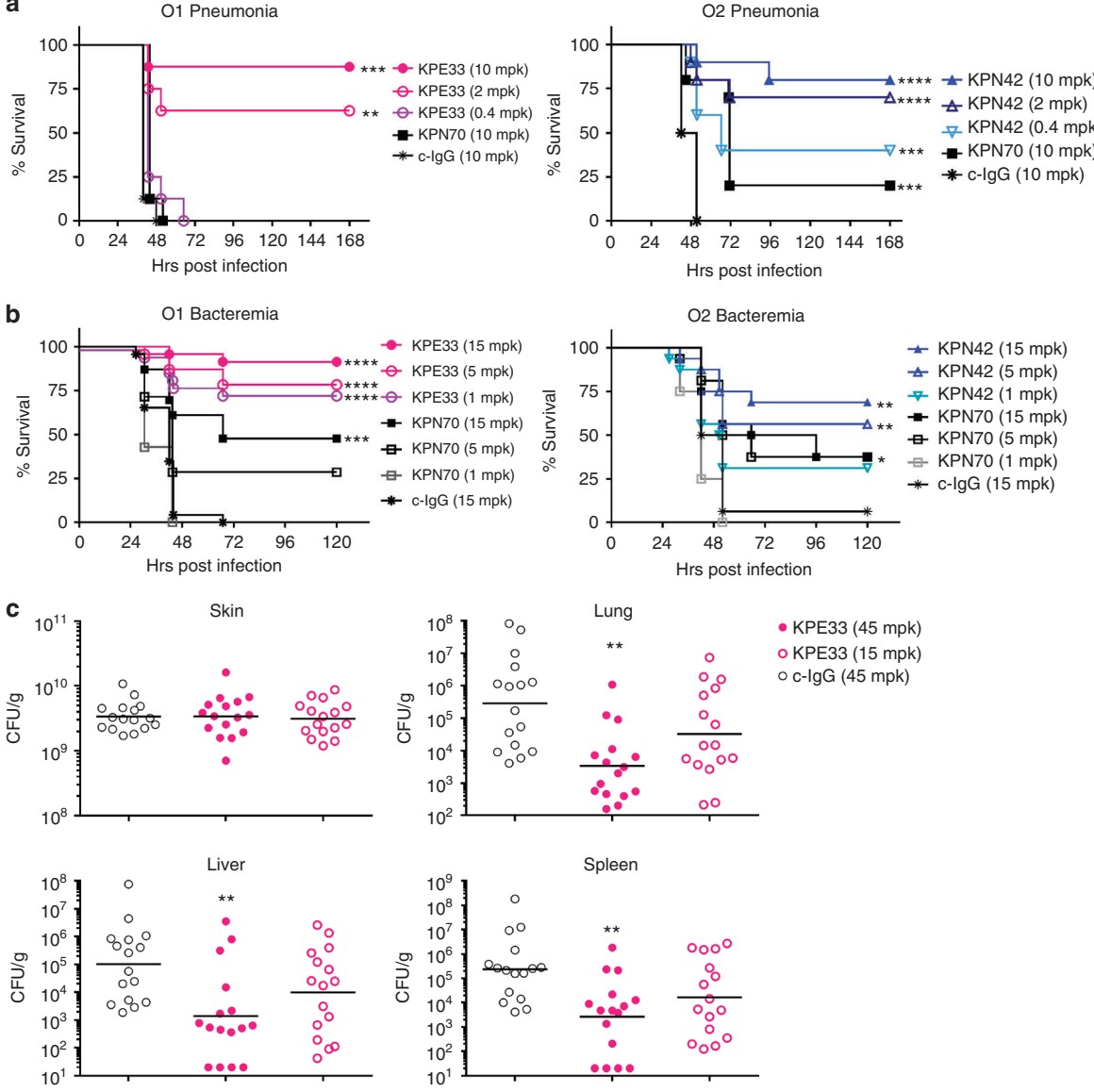

**Fig. 5** Anti-O1 and anti-O2 mAbs are protective in vivo. **a** Therapeutic activity of the mAbs was tested in C57BL/6 mice infected intranasally with a *K. pneumoniae* O1 (Kp1131115, $6 \times 10^7$ CFU) or O2 (Kp961842, $2 \times 10^8$ CFU) strain. KPE33 (anti-O1), KPN42 (anti-O2) and KPN70 (anti-O1/O2) mAbs were administered intravascular 1 h post infection at the doses indicated. An isotype-matched mAb (c-IgG) was used as a negative control. **b** Prophylactic activity of the mAbs was tested in C57BL/6 mice infected intraperitoneal with the O1 strain Kp1131115 ($3 \times 10^6$ CFU) or the O2 strain Kp961842 ($1 \times 10^7$ CFU). mAbs were administered IP 24 h prior to infection at the doses indicated; survival was monitored for 5 days post infection. **c** CF1 mice received a dorsal burn followed by *K. pneumoniae* O1 (Kp1131115, $5 \times 10^6$) bacterial inoculation subcutaneously at the burn site. KPE33 (45 or 15 mpk) or isotype-matched control mAb (c-IgG, 45 mpk) were administered IP 24 h post infection, and organs were harvested 48 h post infection to determine bacterial CFU. All graphs are representative of at least three separate experiments. Mantel-Cox analysis was done to assess significant survival benefit compared to the control mAb. For thermal injury CFU comparison, Kruskal Wallace one way Anova analysis was done to compare differences in CFU between KPE33 treated vs. control IgG treated groups, ****$p < 0.0001$, ***$p < 0.001$, **$p < 0.01$

activity of KPN70 against O1 suggests that LPS neutralization in the absence of opsonophagocytic killing may preferentially provide limited benefit in the bacteremia model vs. the pneumonia model. KPE33 also protected against dissemination to distal organs from necrotic tissue at the burn site in a murine model of thermal injury (Fig. 5c), suggesting the utility of anti-O-antigen mAbs to protect in multiple infection models. Further in vivo testing using additional strains (including Kp8045, a highly virulent, mucoid K1 strain[39]) in a pneumonia model (Supplementary Fig. 5a), confirmed the high level of serotype-specific protection mediated by the KPE33 and KPN42 mAbs as well as

the inferior activity of the cross-reactive, LPS neutralizing mAb KPN70. Interestingly, KPE33 and KPN42 showed in vivo activity even though mAb in vitro activity was restricted to capsule-deficient strains (Fig. 4d, Supplementary Fig. 4). The data from this panel of mAbs, in concert with recently published data showing that an opsonophagocytic killing deficient mutant KPE33 loses activity in vivo[40], supports opsonophagocytic killing activity against specific epitopes as a major mechanism of in vivo protection, at least in the pneumonia model. Conversely, a beneficial role for LPS neutralization may be more limited and dependent on the type of infection.

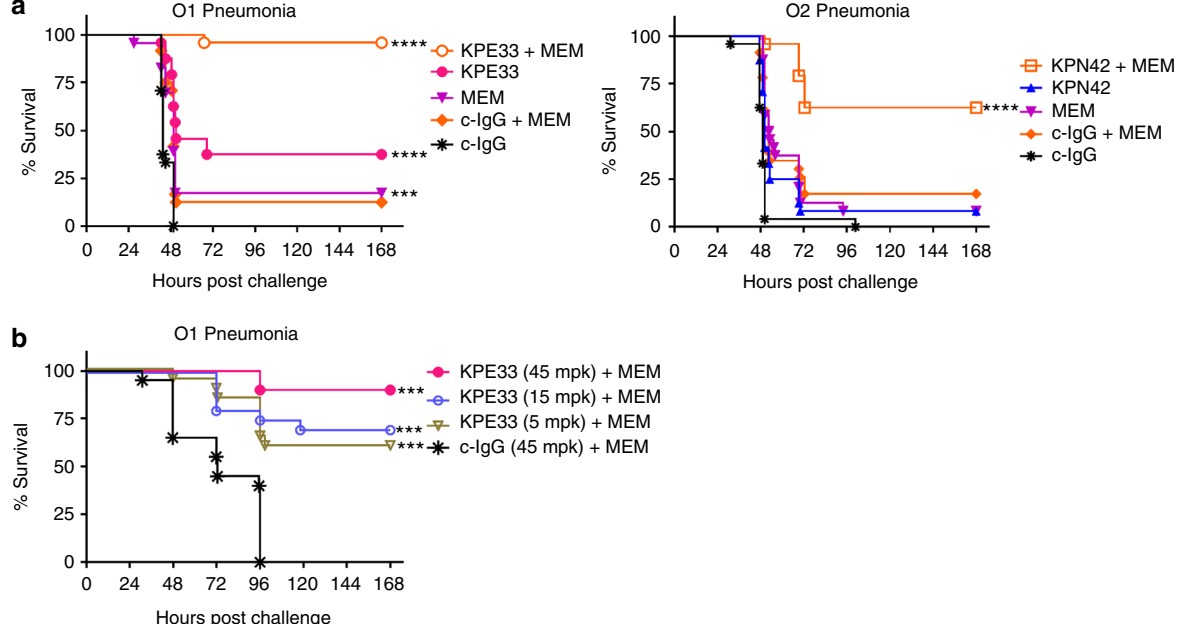

**Fig. 6** Antibody synergy with antibiotic. **a** Mice were infected intranasally as in Fig. 5 with the *K. pneumoniae* O1 (Kp1131115, $6 \times 10^7$ CFU) or O2 (Kp961842, $2 \times 10^8$ CFU) strain then treated 1 h post infection with suboptimal doses of mAb (KPE33, 1 mg kg$^{-1}$ or KPN42, 0.2 mg kg$^{-1}$ given IV) and antibiotic (meropenem, 7.5 mg kg$^{-1}$ for O1 strain; 50 mg kg$^{-1}$ for O2 strain, given subcutaneous) or a combination of both mAb and antibiotic. Graphs represent the combined data from three separate experiments with a total of 24 mice in each group. Mantel-Cox analysis was done to assess significant survival benefit compared to the control mAb (**a**) or control mAb + MEM (**b**), ****$p < 0.0001$, ***$p < 0.001$, **$p < 0.01$. The Bliss independence method was used to determine synergistic activity of the combination therapy vs. either monotherapy alone; $p = 2.5 \times 10^{-6}$ for KPE33+MEM combination and $p = 7.7 \times 10^{-5}$ for KPN42+MEM combination. **b** Mice were infected intranasal with a KP O1 strain (Kp8045, $1 \times 10^4$ CFU). Meropenem (1.5 mpk) was administered subcutaneous 4 h post infection and mAb KPE33 was administered IV 24 h post infection at the indicated doses. All graphs are representative of at least three separate experiments. An isotype-matched mAb (c-IgG) was used as a negative control. Mantel-Cox analysis was done to assess significant survival benefit compared to the control mAb+MEM **b**, ****$p < 0.0001$, ***$p < 0.001$, **$p < 0.01$

**Anti-O-antigen mAbs protection is synergistic with antibiotics.** Clinical application of anti-KP antibodies would likely be in combination with current standard of care antibiotic treatments, including a carbapenem such as meropenem. We therefore assessed the potential for the anti-O1 and O2 mAbs to complement antibiotic therapy using sub-therapeutic doses of antibiotics to simulate insufficient drug exposure against drug-resistant strains, as described previously[10, 11, 41]. These experiments were performed using identical conditions to the pneumonia model described in Fig. 5a with the additional administration of meropenem. A sub-therapeutic dose of mAb KPE33 administered in combination with a sub-protective meropenem dose provided significant protection in the murine pneumonia model against a intermediately meropenem-resistant O1 strain (MIC = 2 µg ml$^{-1}$). Statistical analysis confirmed synergistic adjunctive activity compared to either drug administered alone (Fig. 6a). The efficacy of the mAb was maintained even when administered 24 h post infection (Fig. 6b). Similarly, a sub-therapeutic dose of mAb KPN42 in combination with a dose approximating a human-equivalent level of meropenem, also provided significant protection in this model against a highly meropenem-resistant O2 CRE ST258 strain (MIC = 32 µg ml$^{-1}$, eight times more than the human breakpoint) (Fig. 6a). These data clearly illustrate that mAbs targeting O-antigen could not only provide protection when dosed sufficiently in monotherapy, but could also provide complementary synergistic activity in adjunctive therapy with antibiotics. In conclusion, these results indicate that anti-O-antigen antibodies can provide additional and complementary benefits to antibiotic therapy even against highly carbapenem-resistant KP strains and further substantiate the concept of antibody help in antibiotic therapy[10, 11, 41]. Without this additional antibody-mediated help, O2 strains may preferentially persist under antibiotic pressure.

## Discussion

*K. pneumoniae* is a common enteric bacterial species in human gut flora, therefore basal levels of antibody to immunodominant antigens are to be expected and were indeed evident in our survey of tonsillar and circulating memory B cells. In our efforts to identify protective mAbs against *K. pneumoniae*, only a small fraction of antibodies recognizing whole bacteria possessed potent opsonophagocytic killing activity, the majority of which recognized LPS O-antigen. It is reasonable to consider that humoral immunity to O-antigen may offer some level of protection against the establishment of systemic infections in healthy individuals, and these antigens are currently under clinical investigation as a vaccine target against other pathogens such as nontyphoidal Salmonella, *Vibrio cholera* and *Shigella sonnei*[42–46], though in each case the antibody activity was verified against unencapsulated strains only. Our study determined for the first time that the O2 serotype comprises a significantly larger fraction of ESBL (35%) and CRE (50%) strains in comparison to fully susceptible (17%) KP strains suggesting a selective advantage for MDR strains of the O2 serotype despite their susceptibility to serum killing. Much attention has been focused on the particularly successful MDR carbapenem-resistant ST258 sequence type of *K. pneumoniae*, which we serotyped as almost exclusively O2. However, as ST258 only accounts for half of CRE strains and a much smaller fraction of the earlier emerging ESBL strains, simple clonal outbreaks cannot completely account for the increased prevalence of the O2 serotype.

A possible explanation for the unexpected prevalence of the serum-susceptible O2 serotype is the lower immunogenicity of the O-antigen and the consequent failure to induce antibodies that synergize with antibiotics. Thus, the combination of drug resistance and lack of immunogenicity observed in this study may contribute to the success of O2 strains (including the ST258 subset) particularly in the context of broad spectrum antibiotic pressure. It is notable that strains without a detectable O-antigen (non-serotypeable) were also more prevalent in CRE isolates. This observation further supports the concept that strains with less immunogenic O antigen or lacking O antigen expression may be selected under antibiotic pressure. However, immunogenic O1 strains are also prevalent among ESBL isolates and slightly increased in prevalence compared to their representation in susceptible isolates. In this instance, other O1-specific mechanisms of immune evasion may play a role such as their resistance to human serum killing (Fig. 1b) and/or the increased presence of virulence genes in O1 isolates vs. other serotypes[16]. Human serum sensitivity may play a role in bacterial clearance in patients, but would not necessarily be expected to translate to mouse models as species-specific differences in serum sensitivity have been clearly demonstrated for several pathogens, including *K. pneumoniae*[26–30].

A diminishing pipeline for new antibiotics and the now recognized negative impact of broad-spectrum antibiotics on the beneficial microbiome highlight the necessity of developing new antibacterial strategies[47]. The diversity of capsular polysaccharides of *K. pneumoniae* has been suggested by many to be an immunodominant feature that obscures the accessibility of antibodies to other surface targets. Here, we demonstrate that it is possible to identify highly protective mAbs to the less diverse O-antigen. Despite the relative rarity of O2 antigen-specific B cells, it was possible to identify a mAb (KPN42) with high protective activity. Potent mAbs targeting the O1 antigen were also identified. Interestingly, these mAbs had no opsonophagocytic killing activity against WT strains in vitro. However, in vitro activity against capsule knock-out strains was found to be predictive of activity against WT encapsulated strains in vivo, suggesting capsule may be expressed at lower levels or less consistently in vivo. Consistent with previous studies with other bacterial pathogens[10, 11], the anti-KP mAbs isolated in this study afforded synergistic adjunctive protection to drug-resistant strains, including a representative strain of the problematic ST258 CRE clone. These data further support an under-appreciated role for humoral immunity in antibiotic therapy against KP and indeed the background impact of immunity on bacterial drug resistance observed with vaccines[9]. Furthermore, these studies provide some encouragement that mAb-based strategies may offer future solutions for combatting antibiotic refractory bacteria.

## Methods

**K. pneumoniae strains**. All *K. pneumoniae* isolates were purchased from America Type Culture Collection (ATCC), Eurofin or IHMA (a complete list of strains can be found in Supplementary Table 2), and cultures were maintained in 2xYT media at 37 °C supplemented with antibiotics when appropriate. For capsule mutant strain generation, approximately one kilobase of DNA flanking either side of the *cpsB/manC* gene was PCR amplified and cloned into the pir-dependent plasmid pDMS197[48]. This plasmid was then electroporated into Klebsiella strains and plasmid integrants selected on LB agar containing 10 µg ml⁻¹ tetracycline or gentamycin. After propagation of recombinants in the absence of selection, clones that resolved the integrated plasmid were selected by growth on LB agar containing 10% sucrose. Deletion of *manC* was confirmed by PCR on lysed colonies with primers specific to the *manC* gene (AAACAGTTCCTCCGTCGTTC, GGAA-TAAAGGTGGACTGGTTCT) using the following cycling conditions: 94° for 2 min, 30 cycles of 94° for 30 s, 58° for 30 s, then 72° for 1 min.

**Clinical isolates**. For serotyping, clinical isolates were obtained from International Health Management Associates (Schaumburg, IL, USA). The isolates were selected based on MIC profiles (to ensure relatively equal numbers of susceptible, ESBL and CRE strains), then chosen from 6 continents, 38 different countries and at least 162 different hospitals in an attempt to generate a global snapshot of current *K. pneumoniae* infections as opposed to a sampling of any single outbreak. All isolates were collected between 2012 and 2014 from various sites of infection (a breakdown of infection site can be found in Supplementary Table 1).

**Isolation and production of LPS serotyping antibodies**. Peripheral blood mononuclear cells (PBMC) and sera were separated from buffy coats of healthy blood donors or convalescent patients after *K. pneumoniae* infection as previously described[49]. Alternatively, lymphocytes were obtained from tonsils after tissue homogenization in the presence of DNase I and collagenase. The donors provided written informed consent for the use of these samples, following approval by the Cantonal Ethical Committee of Canton Ticino, Switzerland (for healthy donors) and the Ethical Committee of the Policlinico San Matteo, IRCCS, in Pavia (Italy) (for convalescent ICU donors). Memory B cells were isolated from cryopreserved PMBC or from lymphocytes isolated from tonsils using PE-Cy7-labeled CD19 microbeads (BD Biosciences, clone SJ25C1), followed by isolation with anti PE-beads (Miltenyi Biotec) and by depletion of cells stained with anti-IgM (Jackson ImmunoResearch), anti-IgD (BD Biosciences), and anti-IgA (Novex) antibodies by cell sorting on a FACSAria (BD Biosciences). Memory B cells were immortalized with EBV as described previously[36, 50]. Briefly, sorted memory B cells were infected with EBV and seeded into 384 well plates in the presence of CpG TRL9 agonist and allogenic irradiated PMBCs. Immortalized memory B cells were then screened for antibody binding to purified LPS by ELISA. Total mRNA from positive B cell cultures was reverse transcribed in 50 µl nuclease-free water (Ambion) using 15 µM specific primer (IgG:5′-TCTTGTCCACCTTGGTGTTGCT; Igκ: 5′ACACTCTCCCCTGTT-GAAGCTCTT; Igλ: 5′-ACTGTCTTCTCCACGGTGCT), 1.8 µl of 25 mM dNTP mix (GE Healthcare), 5 µl of 0.1 M DTT, 0.5% v/v Igepal CA-630 (Sigma), 60 U RNAse OUT (Invitrogen), and 100 U Superscript III reverse transcriptase (Invitrogen). Reverse transcription (RT) was performed at 42 °C 10 min, 55 °C 60 min, and 94 °C 5 min. IgH, Igκ, and Igλ genes were amplified by PCR using HotStar Taq DNA polymerase (Qiagen) and cloned into human Igγ1, Igκ, and Igλ expression vectors, as described[51]. Recombinant mAbs were produced by transient transfection of EXPI293 cells (Invitrogen) and purified by Protein A chromatography (GE Healthcare) and desalted against PBS. Alternatively, mouse immune sera were generated by purified LPS or bacterial immunization. The specificity of the resulting mAbs and polysera was confirmed by loading 1 µg of purified LPS from serotype reference strains onto an sodium dodecyl sulfate-polyacrylamide gel electrophoresis (SDS-PAGE), followed by transfer onto a nitrocellulose membrane, and probed using our isolated anti-O-antigen mAbs or polyclonal mouse sera (Supplementary Fig. 1a).

**Enzyme-linked immunosorbent assay**. Spectraplate-384 with high protein binding treatment (custom made from Perkin Elmer) or Nunc 96-well high binding plates were coated overnight at 4 °C with 5 µg ml⁻¹ LPS in phosphate-buffered saline (PBS), pH 7.2, and plates were subsequently blocked with PBS supplemented with 1% BSA (low endotoxin, Sigma-Aldrich). The coated plates were incubated with serial dilutions of our isolated anti-O-antigen human monoclonal antibodies for 1 h at room temperature. The plates were then washed with PBS containing 0.1% Tween-20 (PBS-T), and alkaline phosphatase-affiniPure F(ab′)2 Frag rabbit anti-human IgG, Fcg Frag Specific (Jackson ImmunoResearch, 10,000× dilution) or alkaline phosphatase-goat anti-human IgG (Southern Biotech) were added and incubated for 1 h at room temperature. Plates were washed three times with PBS-T, and P-NitroPhenyl Phosphate (pNPP, Sigma-Aldrich) substrates were added and the absorbance of 405 nm was measured by a microplate reader (Biotek or Molecular Devices). Data were plotted with GraphPad Prism software.

**AMBRA of IgG and IgM antibodies from tonsillar B cells**. Replicate cultures of total tonsillar lymphocytes were seeded at 20,000 cells per well in 96 U-bottom plates (Costar) and stimulated with 2.5 µg ml⁻¹ R848 (3 M) and 1000 U ml⁻¹ human recombinant IL-2 for 10 days at 37 °C 5% CO₂. The cells culture supernatants were collected for further analysis. Spectraplate-384 with high protein binding treatment (Perkin Elmer) or Nunc 96-well high binding plates were coated overnight at 4 °C with 5 µg ml⁻¹ O1 or O2 LPS in phosphate-buffered saline (PBS). Plates were blocked with PBS 1% BSA (low endotoxin, Sigma-Aldrich) and then incubated with AMBRA supernatant for 1 h at room temperature. After washings with PBS 0.1% Tween the binding of IgG or IgM Abs was detected with goat Alkaline Phosphatase-anti human IgG (Southern Biotech) or Alkaline Phosphatase-goat anti human IgM (Southern Biotech), respectively. pNPP (Sigma-Aldrich) substrate was added and absorbance measured at 405 nm on a microplate reader (Biotek).

**Serotyping of K. pneumoniae clinical isolates**. The serotypes of the KP clinical isolates were determined by western blot analysis with serotype-specific monoclonal antibodies or mouse polysera. Briefly, purified LPS or bacterial lysates were subjected to SDS-PAGE. Separated proteins and LPS were transferred from gels to nitrocellulose membranes with an iBlot apparatus (Life Technology). Membranes were then blocked with Casein or Odyssey (Li-cor) blocking buffer before being probed with anti-O1, O1/O2, O3, O4, or O5 LPS monoclonal antibodies or anti-O7

and anti-O12 mouse polysera. After repeated washes with PBS-T, blots were incubated with IRDye680 or 800 fluorescent secondary antibodies (Li-cor) and visualized with an Odyssey Image Station (Li-cor). In some circumstances, bacterial lysates were treated with 0.4 mg ml$^{-1}$ Proteinase K (ThermoScientific) to remove protein components before the western blot analysis. CRE isolates were defined as resistant to carbapenems (doripenam, imipenam and/or meropenam MIC > = 4 µg ml$^{-1}$), ESBL isolates defined as resistant to cephalosporins (ceftazidime MIC > = 16 µg ml$^{-1}$) but sensitive to carpapenams (MIC < 4 µg ml$^{-1}$) and susceptible isolates defined by ceftazidime MIC < 16 µg ml$^{-1}$ and carbapenam MIC < 4 µg ml$^{-1}$.

**Octet binding assay with anti-LPS mAbs.** Anti-LPS mAbs diluted to 0.2 µg ml$^{-1}$ in PBS were immobilized for 10 min at 37 °C on the surface of a protein A coated sensor-chip of an Octet RED96 system (FortéBio). Coated-Biosensors were incubated for 10 min with a solution containing 2 µg ml$^{-1}$ O1 or O2 LPS (purified from Kp43816ΔcpsB or Kp8570ΔcpsB, respectively) in Kinetics buffer (ForteBio, dilute 10× to 1× with PBS). A dissociation step was then performed incubating the Biosensor for 20 min in Kinetics buffer. Changes in the number of molecules bound to the biosensor caused a shift in the interference pattern that was recorded in real time and off-rate analyses were performed to determine mAbs KD (M).

**Blockade of binding assay.** Human anti-LPS mAbs were biotinylated using the EZ-Link NHS-PEO solid phase biotinylation kit (Pierce). Labeled mAbs were tested for binding to LPS by ELISA and the optimal concentration of each mAb to achieve 70% maximal binding was determined. Unlabeled mAbs were serially diluted and added to ELISA 96-well plates (Corning) coated overnight at 4 °C with 5 µg ml$^{-1}$ LPS in PBS. After 1 h, biotinylated anti-LPS mAbs were added at the concentration achieving 70% maximal binding and the mixture was incubated at room temperature for 1 h. Plates were washed and antibody binding was revealed using Alkaline Phosphatase-streptavidin (Jackson Immunoresearch). After washing, pNPP substrate (Sigma-Aldrich) was added and plates were read at 405 nm. The percentage of inhibition was calculated as a percent decrease of signal vs. the maximum signal achieved with the labeled mAb.

**MLST analysis.** Multi-locus sequence type (MLST) analysis was performed using the method and primers as described elsewhere[52]. Briefly, a bacterial colony was resuspended in 40 µl DNase free water and used as template for PCR reactions using primers specific to each of the seven MLST genes with the inclusion of a universal 3′ sequence tag on each primer (**rpoB** forward GTTTTCCCAGTCAC-GACGTTGTA**GGCGAAATGGCWGAGAACCA**, reverse TTGTGAGCGGA-TAACAATTTC**GAGTCTTCGAAGTTGTAACC**; **gapA** forward GTTTTCCCAGTCACGACGTTGTA**TGAAATATGACTCCACTCACGG**, reverse TTGTGAGCGGATAACAATTTC**CTTCA-GAAGCGGCTTTGATGGCTT**; **mdh** forward GTTTTCCCAGTCAC-GACGTTGTA **CCCAACTCGCTTCAGGTTCAG**, reverse TTGTGAGCGGATAACAATTTC**CCGTTTTTCCCCAGCAGCAG**; **pgi** forward GTTTTCCCAGTCACGACGTTGTA**GAGAAAAACCTGCCTGTACTGCTGGC**, reverse TTGTGAGCGGATAACAATTTC**CGCGCCACGCTTTA-TAGCGGTTAAT**; **phoE** forward GTTTTCCCAGTCACGACGTTGTA**ACC-TACCGCAACACCGACTTCTTCGG**, reverse TTGTGAGCGGATAACAATTTC**TGATCAGAACTGGTAGGTGAT**; **infB** forward GTTTTCCCAGTCACGACGTTGTA**CTCGCTGCTGGACTATATTCG**, reverse TTGTGAGCGGATAACAATTTC **CGCTTTCAGCTCAAGAACTTC**; **tonB** forward GTTTTCCCAGTCACGACGTTGTA**CTTTATACCTCGGTA-CATCAGGTT**, reverse TTGTGAGCGGATAA-CAATTTC**ATTCGCCGGCTGRGCRGAGAG**). After exonuclease clean up using Exo-SapIT (Affymetrix), the resulting template was used for a BigDye Terminator (Affymetrix) PCR reaction according to the manufacturer's protocol using universal primers for all 7 genes (forward GTTTTCCCAGTCACGACGTTGTA, reverse TTGTGAGCGGATAACAATTTC) followed by analysis on the ABI 3730 (Applied Biosystems). SeqScape software (ABI) was used to match sequence data to the public database (http://bigsdb.pasteur.fr/klebsiella/klebsiella.html) and to determine the ST of each strain. In some cases, assembled whole genome sequences were used to determine ST type.

**Opsonophagocytic killing assays.** Opsonophagocytic killing activity of anti-O antigen mAbs was tested against O1 and O2 strains. Briefly, log phase cultures of luminescent KP strains 8570ΔcpsBlux (O2) and 43816ΔcpsBlux (O1) were diluted to ~2 × 10$^6$ cells ml$^{-1}$. Bacteria, 5 × 10$^5$ dimethylformamide (DMF) differentiated HL-60 cells, 1:10 diluted cleared baby rabbit serum (Cedarlane), and a series dilution of antibodies (2 ng–2500 ng m l$^{-1}$) were mixed in 96-well plates and incubated at 37 °C for 2 h with shaking (250 rpm). The relative light units (RLUs) were then measured using an Envision Multilabel plate reader (Perkin Elmer). The percent killing was determined by comparing RLU derived from assays with no antibodies to RLU derived from assays with anti-KP or negative control mAbs.

**Confocal imaging.** K. pneumoniae strains were inactivated with formaldehyde, washed and re-suspended in PBS to achieve an OD$_{600}$ of 3. In each experimental condition, 50 µl of bacterial suspension were centrifuged and re-suspended in 5% FBS in PBS. The tested mAbs were added to the bacterial suspension at a final

concentration of 10 µg ml$^{-1}$ and incubated for 2.5 h at 4 °C. After two wash steps, fluorescent anti-human IgG secondary antibody (Jackson Immunoresearch) and DAPI were added to the pellet to a final concentration of 15 µg ml$^{-1}$ and 4 µg ml$^{-1}$, respectively, and incubated for 40 min at 4 °C. Bacteria were washed twice with PBS, resuspended in a Mowiol-based mounting medium and mounted on microscopy slides. Slides were air-dried O/N in the dark and images were taken using the TCS SP5 microscopy system from Leica using a ×100 objective and confocal settings.

**LPS neutralization assays.** A murine RAW264.7 macrophage cell line was engineered to carry a firefly luciferase reporter gene under the control of an NF-κB-responsive promoter (RAW264.7-lux). Serially diluted antibody stocks were mixed with purified KP LPS (2 ng ml$^{-1}$ final concentration) and incubated at 4 °C for 1 h. Antibody/LPS mixtures were then added to assay plates containing pre-seeded RAW264.7-lux cells (5000 cells per well). Plates were incubated for 2.5 h at 37 °C with 5% CO$_2$ then Steady Glo solution (Promega) was added to each well and incubated for another 20 min protected from light. The RLUs were measured using an Envision Multilabel plate reader (Perkin Elmer). The percentage of inhibition was determined by comparing RLU derived from assays with no antibodies to RLU derived from assays with anti-KP or negative control mAbs.

**Bacterial infection models.** C57BL/6 mice were received from Jackson laboratories and maintained in a special pathogen free facility. All animal experiments were conducted in accordance with IACUC protocol and guidance. K. pneumoniae strains were grown on agar plates overnight and diluted in saline at proper concentration just prior to infection. The inoculum titer was determined by plating a serial dilution of bacteria onto agar plates. Mice were inoculated with 5 × 10$^3$ to 2 × 10$^8$ CFU of KP clinical isolates either intranasally (pneumonia model) or intraperitoneally (bacteremia model). Anti-LPS monoclonal antibody and human IgG1 control antibody were administered IV 1 h post-bacterial challenge (therapeutic dosing) or IP 24 h prior to infection (prophylactic dosing). Mouse survival was monitored daily. For the thermal injury model, CF1 mice (Charles River Laboratories) received a dorsal burn under anesthetic for 5 s followed by administration of 5 × 10$^6$ bacteria subcutaneously at the burn site. MAb was administered IP 24 h post infection, organs were harvested at 48 h post infection and bacterial CFU was determined by serial dilution of organ homogenate. For antibody and antibiotic combination studies, both were given 1 h post infection (mAb delivered IV, meropenem delivered subcutaneously at 7.5 mg kg$^{-1}$ in O1 model or 50 mg kg$^{-1}$ in O2 model). For extended timecourse experiments, meropenem was delivered 4 h post infection and mAb 24 h post infection. Survival data was plotted in Prism and statistical analysis was determined using the Log-rank Mantel-Cox test. To determine synergy in the antibody/antibiotic combination experiments, the Bliss independence method for drug combination synergy test was used as described elsewhere[53]. Briefly, if the rate of survival at the end of study for drug A alone is $r_a$ and the survival rate for drug B alone is $r_b$, then the expected survival rate for drug A and drug B combination is $r_{Bliss} = r_a + r_b - r_a r_b$ assuming that the two drugs are bliss independent. The difference between the observed survival rate $r_{ab}$ and the expected rate is defined as the synergy index.

**Ethics statement.** The donors provided written informed consent for the use of blood samples, following approval by the Cantonal Ethical Committee of Canton Ticino, (Switzerland), for healthy donors and the Ethical Committee of the Policlinico San Matteo, IRCCS, in Pavia (Italy) for convalescent and intensive care donors. All animal studies were performed under the guidance and protocol approval of the MedImmune Institutional Animal Care and Use Committee. Additional oversight was also provided by Office of Research Protections (ORP), US Army Medical Research and Material Command (USAMRMC), Animal Care and Use Review Office (ACURO).

**Data availability.** The authors declare that all relevant data supporting the findings of the study are available in this article and its Supplementary Information files, or from the corresponding author upon request.

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

## Acknowledgements

This work was funded by MedImmune, LLC, a wholly owned subsidiary of AstraZeneca Pharmaceuticals and a grant from the Defense Advanced Projects Research Agency (DARPA). We thank Professor Piero Marone from Policlinico San Matteo in Pavia, Italy for providing the sera samples from convalescent patients.

## Author contributions

M.E.P., Q.W., P.W., J.S., D.C., E.C., A.D.M., M.B., A.L., and C.K.S. contributed to the experimental design and writing of this manuscript. M.E.P., Q.W., M.P., A.D.M., J.B., R. C., M.B., X.X., E.C., W.Z., M.M.C., S.B., F.Z., A.D., and E.S. performed experiments or provided critical reagents.

## Additional information

**Competing interests:** M.E.P., Q.W., M.P.,J.B, R.C, E.S, X.X, W.Z., M.M.C, A.D., P.W., J.S., C.K.S. are current or former employees of MedImmune/AstraZeneca and may own stock or stock options in the company. Patents describing the activity of the antibodies in this work have been filed by MedImmune. A.D.M., M.B, E.C., S.B, F.Z, A.L, D.C. are employees of HumAbs BioMed and may currently hold Humabs stocks or stock options. The remaining authors declare no competing financial interests.

