## [Peer Review File · Nature Communications]

Editorial Note: Parts of this peer review file have been redacted as indicated to remove third-party material where no permission to publish could be obtained.

Reviewers' comments:

Reviewer #1 (Remarks to the Author):

Novel therapeutic strategies to target the ST259 *K. pneumoniae* are clearly needed and the development of highly active mAbs could be useful. The data provided in this manuscript are intriguing and clearly delineate a major challenge, namely the lack of immunogenicity of the LPS associated with a substantial number of these strains. While the data documenting the prevalence of the O2 and other serotypes is straightforward and the demonstration of the lack of immunogenicity of this serotype is convincingly demonstrated, the utility of this epitope in the development of mAbs is less convincing. Several questions are not directly addressed and could help to clarify the importance of the findings.

1 – The authors demonstrate a lack of correlation between serum sensitivity, LPS neutralization and outcome. Are the D-galactan I and II moieties expressed in the ST258 capsule as well as the LPS? Since LPS seems not to be the critical factor, in contrast to opsonophagocytic killing (OPK should be defined), does Ab affect uptake, internalization or mediate killing by the phagocytes? Does this explain its activity? If the O2 strains are susceptible to complement, how long does bacteremia persist? Is clearance by the Ab TLR4 dependent or are other pathways involved.

2- In contrast to the data provided to characterize Ab production and opsonophagocytosis, the *in vivo* efficacy studies in fig 5 are difficult to follow. Different strains of O1 and O2 types were used, different inocula, different routes and times of Ab administration (24 hours pre-infection, 1 hour post – IV versus *i. p* ?). The lack of uniformity in the methods makes it exceedingly difficult to interpret the results. Although the virulence of the different strains is clearly an issue, lethality at 48 hours in the controls does provide a “standardization” that should enable comparison of the Abs at identical doses and routes of administration. Given the high degree of mortality, it is likely that the pneumonia model causes bacteremia. Why wasn't KPN42 more efficacious in the bacteremia model? Was this just due to route or timing of administration? Similarly the data with combined use of Ab and meropenem (fig 6) use yet another set of conditions, and seem to neglect to show the inoculum. As mice treated with the mAb alone in fig 5 had some survival, it is curious that without the antibiotic (in fig 6) but with 50 mg/kg of the KPN42, all succumbed by 72 hours. This is not a convincing demonstration that the targeting of the O2 antigen is especially efficacious.

Reviewer #2 (Remarks to the Author):

Antibiotic resistance is a major public health threat especially when manifested by *Klebsiella pneumoniae* (KP). KP is now one of the most concerning multidrug resistant pathogens. These investigators report a striking increased prevalence of KP lipopolysaccharide (LPS) O2 serotype strains in all major drug resistance groups which may be explained by a marked paucity of anti-O2 antibodies in human B cell repertoires. They also identify human monoclonal antibodies to O-antigens that are highly protective in mouse models of infection even against heavily encapsulated strains. These antibodies, including a rare anti-O2 specific antibody, also synergistically protect against drug resistant strains in adjunctive therapy with meropenem, a standard-of-care antibiotic, confirming the importance of immune assistance in antibiotic therapy. These findings support a viable antibody based therapeutic strategy to combat even highly resistant KP infections and also underscore the impact of humoral immunity on evolving drug resistance. These findings are important as they point the way to a novel therapeutic approach.

Reviewer #1

1. The authors demonstrate a lack of correlation between serum sensitivity, LPS neutralization and outcome. Are the D-galactan I and II moieties expressed in the ST258 capsule as well as the LPS?

The reviewer is correct in pointing out that LPS and capsule can share similar sugar components. However, the anti-O2 mAb KPN42 described in this study does not bind the ST258 strain 961842 when the O2 antigen is converted to O1, as shown in Fig. S6B. This result confirms that KPN42 does not cross-react with the ST258 capsule that is shared between the parental WT and mutant strains. To further support this conclusion, we generated additional O-antigen deletion mutants (961842 Δ wecA and 1131115 Δ wecA) to show that all of our mAbs (KPE33, KPN42 and KPN70) only bind to WT strains but not their isogenic O-Ag deficient mutants with complete capsules. This new data set was added to Fig. S6C along with additional text (lines 194-196) and further supports that our O-antigen specific mAbs target distinct O-antigen epitopes not shared by capsular polysaccharides.

Since LPS seems not to be the critical factor, in contrast to opsonophagocytic killing (OPK should be defined), does Ab affect uptake, internalization or mediate killing by the phagocytes? Does this explain its activity?

The reviewer underlines the importance of the OPK activity of the mAbs compared to LPS neutralization. Recent data from our group supporting OPK as the critical mechanism of action was reported in *JCI Insight*. 2017;2(9):e92774 as now referenced in the text. A N297Q mutation was introduced in the mAb Fc region which inhibits the interaction of antibody with phagocyte Fc receptors^{2,3}. The N297Q mutant mAb, which is deficient for in vitro OPK activity, does not significantly improve survival in KP infected mice and bacterial burden in the lung is significantly higher in mice treated with the mutant mAb compared to WT KPE33 (*JCI Insight*, 2017, Fig. 2E and S5B, inserted below for Reviewers only). Taken together these data indicate that both uptake and bacterial killing are critical to the function of anti-O-Ag mAbs, while potent LPS neutralization is less important or may be even counterproductive by eliminating stimulation of immune effector cells, at least in the pneumonia model. As suggested by the Reviewer, we have now better highlighted this reference in the text to support OPK activity as a critical mechanism of action (lines 238-241).

[Redacted]

If the O2 strains are susceptible to complement, how long does bacteremia persist?

As we introduced in the results, the relative serum resistance of the O1 strain has been previously cited as a possible explanation for the O1 serotype's relatively dominant prevalence. We therefore wanted to address the possibility that O2 strains may also have a competitive advantage due to serum resistance. However, we found that despite the increase in O2 prevalence in multiple classes of drug resistant strains, O2 strains are indeed serum sensitive (Fig. 1B). We therefore believe serum resistance is likely not as important as generally believed and that other virulence factors such as O2 immune stealth (a major point in this work) are potentially more important. As both Reviewers agree, the lack of O2 immunogenicity in multiple animal species and in humans is striking. Though O2 strains are clearly more susceptible to human serum (Fig. 1B), there are several examples in the literature that demonstrate bacterial growth inhibition in human serum is not replicated in mouse serum, including a recent publication in which KP ST258 strains (O2 serotype) were extremely sensitive to human, but not mouse serum^{4, 5, 6, 7, 8}. The O2 strains used in our studies required higher doses to achieve full lethality in mice (for both pneumonia and bacteremia infections) which might be attributed to an increased susceptibility to complement but are more likely due to other factors, such as capsule type. Neither O1 nor O2 infections persist after 72 hours (full lethality is achieved in both strains). We have added text (lines 95-96, 99, 301-304) to better clarify that our human serum data address past explanations of O1 prevalence and is a human observation that would not be expected to translate to the mouse model.

Is clearance by the Ab TLR4 dependent or are other pathways involved.

We agree that examining TLR4 dependence is a very interesting experiment to address the pathways involved in Ab-mediated protection. In the previously mentioned *JCI Insight* article, we showed KPE33 has decreased activity in mice deficient in TLR4 signaling (C3H/HeJ) versus WT mice (¹*JCI Insight*, 2017, Fig. S2). This result from our published companion paper is shown below and indicates LPS-induced TLR4 signaling is required for optimal mAb protective activity. In the *JCI Insight* article we further investigated the role of LPS neutralization early in infection and found that it may be detrimental to KP clearance due to the blocking of beneficial engagement of TLR4 by LPS.

[Redacted]

2. In contrast to the data provided to characterize Ab production and opsonophagocytosis, the in vivo efficacy studies in fig 5 are difficult to follow. Different strains of 01 and 02 types were used, different inocula, different routes and times of Ab administration (24 hours pre-infection, 1 hour post – IV versus i. p ?). The lack of uniformity in the methods makes it exceedingly difficult to interpret the results. Although the virulence of the different strains is clearly an issue, lethality at 48 hours in the controls does provide a “standardization” that should enable comparison of the Abs at identical doses and routes of administration.

The Reviewer is concerned with the use of different conditions in different infection models. We regret their confusion and have significantly modified the text and performed additional experiments to normalize our models. Our goal was to demonstrate that the mAbs are effective against multiple strains of KP (to reflect the heterogeneity found in clinical isolates) and against multiple models of infection. The efficacy of the anti-O1 (KPE33) and anti-O2 (KPN42) mAbs are not directly comparable because of their specificity for different serotypes. Though they both target LPS, they clearly each target unique and distinct O-antigens (see Fig. 4) and are not expected to behave identically. In addition, each model was developed to emphasize different aspects of clinical Klebsiella infection and were not designed to be normalized to each other.

The Reviewer is correct in assuming that bacteria in the pneumonia model eventually disseminate systemically, but this usually takes at least 48 h (depending on the strain used). Therefore, the treatment window is expected to be extended in this model versus the bacteremia model in which bacteria are introduced IP and are immediately disseminated systemically (concurrent with infection). The bacteremia model is a more acute, drastic infection in which the lethal dose is lower compared to pneumonia using the same strain. In light of those observations, we chose to administer mAbs prophylactically in bacteremia as it represents a model of late stage systemic infection. In the pneumonia model, mAbs were administered in treatment 1 h post because this mimics an early infection which is not yet systemic, leaving space for a treatment window. However, the mAbs also work when administered prophylactically in the pneumonia model as reported in the JCI article (the relevant figure is shown above).

To improve the strength of our conclusions and to make the data as clear as possible to the reader, we performed additional experiments in which the mAbs were administered therapeutically in bacteremia at the same time point and route of administration as in our pneumonia model (Fig. S5B). The mAbs still provided significant survival benefit when administered post-infection though they were not as potent as when given prophylactically (Fig. 5B). This result was expected due to the additional systemic burden that occurs rapidly in the bacteremia model. Text was added to highlight similarities and differences in models and dosing (lines 214-216, 218-222, 226-229, Fig. 5 and S5 legends and Methods). Moreover, additional experiments were performed to keep the strains, bacterial challenge dose and mAb doses consistent throughout Fig. 5. This figure now has data using a single O1 strain and a single O2 strain and the doses of mAbs are consistent between experiments. The data regarding different strains and delivery of mAb prophylactically for bacteremia are now reported in Figure 5B and Supplementary Figure 5.

Given the high degree of mortality, it is likely that the pneumonia model causes bacteremia. Why wasn't KPN42 more efficacious in the bacteremia model? Was this just due to route or timing of administration?

The Reviewer is concerned that KPN42 was not more effective in the bacteremia model. It's true the bacteremia model is systemic and therefore more difficult to overcome (less bacteria are required to cause lethality) which may explain differences in the efficacy of the mAbs versus the pneumonia model. However, as shown in Figure 5, the doses required to protect in each of the models are comparable. KPN42 at 15 mpk generates nearly 75% survival, similar to the survival levels seen in the pneumonia model with the same mAb dose. We agree with the Reviewer that the timing of administration of the mAbs can make a critical difference in their ability to protect and have now added data to show that mAbs are more potently protective when delivered prophylactically in the bacteremia model (Fig. S5B). However, IV dosing of the mAb post infection also significantly protected mice in the bacteremia model (Fig. S5B). The mAbs described in this study clearly protect under all the various conditions tested supporting the notion that these anti-O-antigen mAbs have the potential to provide clinical benefit against various types of infection.

Similarly the data with combined use of Ab and meropenem (fig 6) use yet another set of conditions, and seem to neglect to show the inoculum. As mice treated with the mAb alone in fig 5 had some survival, it is curious that without the antibiotic (in fig 6) but with 50 mg/kg of the KPN42, all succumbed by 72 hours. This is not a convincing demonstration that the targeting of the O2 antigen is especially efficacious.

The Reviewer is concerned with the antibiotic plus antibody pneumonia experiment shown in Fig. 6 not being consistent with the pneumonia in the absence of antibiotic experiment in Fig. 5. The timing,

bacterial inoculum and mAb doses used are consistent in each experiment and differed only by the addition of the antibiotic. We apologize for the confusion and appreciate the opportunity to revise the text to better highlight the experimental conditions used. In order to measure the additional benefit of antibiotic, it is important to choose a dose of mAb that does not give significant protection alone. We did not administer 50 mg/kg of KPN42 in Fig. 6 as indicated by the reviewer (we agree that would be extremely inconsistent with Fig. 5). In Fig 6, meropenem was given at 50 mg/kg, while KPN42 was delivered at the non-effective dose of 0.2 mpk (based on Fig. 5 data showing there was still a survival benefit using 0.4 mpk of KPN42). The route and dose of bacteria administered (6×10^7 CFU for 1131115, 2×10^8 CFU for 961842 in both figures, all delivered IN), route of administration and timing for the mAbs (IV given 1 h post infection) are identical between the 2 graphs. We have added text to the Figure legends to clarify that point. In Figure 6C, the mAb was delivered IV at a later timepoint (24 h post infection) to demonstrate the potential efficacy of the mAb even when delivered late during infection. We appreciate the opportunity to clarify and expand our data describing the efficacy of the mAbs in various models of infection and using various routes of administration.

Reviewer #2

Antibiotic resistance is a major public health threat especially when manifested by *Klebsiella pneumoniae* (KP). KP is now one of the most concerning multidrug resistant pathogens. These investigators report a striking increased prevalence of KP lipopolysaccharide (LPS) O2 serotype strains in all major drug resistance groups which may be explained by a marked paucity of anti-O2 antibodies in human B cell repertoires. They also identify human monoclonal antibodies to O-antigens that are highly protective in mouse models of infection even against heavily encapsulated strains. These antibodies, including a rare anti-O2 specific antibody, also synergistically protect against drug resistant strains in adjunctive therapy with meropenem, a standard-of-care antibiotic, confirming the importance of immune assistance in antibiotic therapy. These findings support a viable antibody based therapeutic strategy to combat even highly resistant KP infections and also underscore the impact of humoral immunity on evolving drug resistance. These findings are important as they point the way to a novel therapeutic approach.

We thank the Reviewer for the positive comments and we are pleased that the Reviewer recognizes the therapeutic value of our antibodies. The CDC recently identified drug resistant *Klebsiella* as a Priority 1 pathogen and we agree that finding novel therapeutic approaches is absolutely critical.

1. Cohen T, Pelletier M, Cheng L, Pennini M, Bonnell J, Cvitkovic R, *et al.* Anti-LPS antibodies protect against *Klebsiella pneumoniae* by empowering neutrophil-mediated clearance without neutralizing TLR4. *JCI Insight* 2017.
2. Tao MH, Morrison SL. Studies of aglycosylated chimeric mouse-human IgG. Role of carbohydrate in the structure and effector functions mediated by the human IgG constant region. *J Immunol* 1989, **143**(8): 2595-2601.

3. DiGiandomenico A, Keller AE, Gao C, Rainey GJ, Warrener P, Camara MM, *et al.* A multifunctional bispecific antibody protects against *Pseudomonas aeruginosa*. *Science translational medicine* 2014, **6**(262): 262ra155.
4. Brown GC. The complementary activity of mouse-serum. *J Immunol* 1943, **46**(5): 319-323.
5. Muschel LH, Muto T. Bactericidal reaction of mouse serum. *Science* 1956, **123**(3185): 62-64.
6. Warren HS, Fitting C, Hoff E, Adib-Conquy M, Beasley-Topliffe L, Tesini B, *et al.* Resilience to bacterial infection: difference between species could be due to proteins in serum. *The Journal of infectious diseases* 2010, **201**(2): 223-232.
7. Siggins MK, Cunningham AF, Marshall JL, Chamberlain JL, Henderson IR, MacLennan CA. Absent bactericidal activity of mouse serum against invasive African nontyphoidal *Salmonella* results from impaired complement function but not a lack of antibody. *J Immunol* 2011, **186**(4): 2365-2371.
8. Szijarto V, Guachalla LM, Hartl K, Varga C, Badarau A, Mirkina I, *et al.* Endotoxin neutralization by an O-antigen specific monoclonal antibody: A potential novel therapeutic approach against *Klebsiella pneumoniae* ST258. *Virulence* 2017: 1-13.

REVIEWERS' COMMENTS:

Reviewer #1 (Remarks to the Author):

To the authors –

The revised manuscript is substantially improved. There is a tremendous amount of data provided, that was somewhat difficult to appreciate in the original manuscript. The repeated references to the “companion” JCI Insight paper in the response to the Reviewers implies that most readers will have a familiarity with the authors’ previous work, that may or may not be the case. The revision is much easier to follow and makes fewer assumptions. In particular, the mouse protection studies, which were rather murky, are now clearly presented and the conclusions drawn are justified. This is a nice contribution.

REVIEWERS' COMMENTS:

Reviewer #1 (Remarks to the Author):

To the authors –

The revised manuscript is substantially improved. There is a tremendous amount of data provided, that was somewhat difficult to appreciate in the original manuscript. The repeated references to the “companion” JCI Insight paper in the response to the Reviewers implies that most readers will have a familiarity with the authors’ previous work, that may or may not be the case. The revision is much easier to follow and makes fewer assumptions. In particular, the mouse protection studies, which were rather murky, are now clearly presented and the conclusions drawn are justified. This is a nice contribution.

The Reviewer was satisfied with the revisions we made and did not have any additional requests.